# Sparse Mamba: Reinforcing Controllability In Structural State Space Models

## Abstract

In this work, we introduce the concept of controllability and observability to the Mamba SSM's architecture in our Sparse-Mamba (S-Mamba) for natural language processing (NLP) applications. The structured state space model (SSM) development in recent studies, such as Mamba and Mamba2, outperformed and solved the computational inefficiency of transformers and large language models at small to medium scale. The Mamba SSMs architecture drops the need for attention layers or multilayer perception blocks in transformers. However, current Mamba models lack reinforcement of controllability in state-space equations for computing the $A$, $B$, $C$, and $D$ matrices at each time step, leading to increased complexity and computational costs. In this paper, we demonstrate a reduction of parameters in comparison to the first published Mamba and Mamba2. We showcase an improvement in perplexity by 5% and a decrease in training time by 3% after reinforcing controllability and observability on the original Mamba architecture in our proposed S-Mamba. The controllable $n \times n$ state matrix $A$ is sparse and it has only $n$ free parameters. Our novel approach will ensure a controllable system which will be the gate key for Mamba3.

## 1 Introduction

**Transformers.** In the early stages of natural language processing (NLP), with one of its first studies Hutchins (2005), recurrent neural networks (RNNs) Rumelhart et al. (1986) suffered from exploding/vanishing gradients. This case was investigated by Hochreiter in Hochreiter (1998), first discussed in his thesis in 1991. This study explored four types of solutions including methods which do not use gradients, ones that keep gradients on larger values, ones that operate on higher levels, and ones that use special architectures. This inspired the creation of a gradient-based method Long short-term memory (LSTM) in Hochreiter & Schmidhuber (1997), where constant error carrousel was introduced.

The long sequences in language modeling and generating in an encoder-decoder based architectures as in RNNs and Generative Adversarial Nets (GANs) Goodfellow et al. (2014) was a main problem. In Vaswani (2017), authors revolutionized NLPs with their introduction of transformers. Attention mechanism was all you need to handle long sequences. The core of a transformer model relays in the proposed attention equation. Here, $Q$, $K$, and $V$ are the query, keys values matrices. $W^Q, W^K, W^V$ are projection matrices for the queries, keys, and values, respectively. $W^O$ is the output projection matrix. When these matrices are properly calculated, they form a similarity score in the attention layer that handles longer language modeling tasks more effectively. Further development on transformers produced multi-query attention Shazeer (2019) and flash attention (Dao et al. (2022),Dao (2023)).

**State Space Models (SSMs).** When state space models are discussed, it is often referred to the state space representation and classical state space models introduced by Kalman (1960). Recently, studies attempted to build upon the state space representations in control theory and modeling a dynamic system via state variables to language modeling came as in Gu et al. (2021a). However, in order to make a bridge to language modeling from state space representations, Gu in Gu et al. (2021b) experimented the first known utilization of state space equations appeared as a Linear State-Space Layer (LSSL), where the LSSL maps a sequence input to output using state space equations. Unsurprisingly, similar to transformers, these attempts were also inspired by RNNs. In Gu et al.

(2022), authors proposed a diagonal structure to the first state space model called S4. This S4 model, and previously LSSLs, was built on core state space representation discussed in control theory literature as in Hangos et al. (2006).

Authors of S4 left the idea of expanding the SSM in the coefficient space and started computing its truncated generating function in frequency space. The parameter $D$ was also omitted by setting it to $D = 0$ as it only worked as a skip-connection. A convolution kernel $\overline{K}$ was introduced as non-circular convolution that can be computed very efficiently using FFTs. This will be discussed more in section [2].

However, the fundamental challenge in sequence modeling is compressing context into a smaller state. Popular sequence models, such as Transformers, recurrent models and recent SSMs, illustrate this trade-off. Attention-based models, like Transformers, are highly effective but inefficient because they avoid compressing context, requiring the storage of the entire sequence during inference, leading to slow, quadratic-time training. Recurrent models and S4, while more efficient with constant-time inference and linear-time training, struggle with effectiveness due to limited context compression. This challenge is highlighted by tasks like Selective Copying Arjovsky et al. (2016), which requires filtering relevant tokens, and Induction Heads Olsson et al. (2022), which demands context-aware output generation. These tasks reveal the limitations of linear time-invariant (LTI) models, as they lack the capacity for input-dependent dynamics and struggle with varying input-output spacing, a problem static convolution kernels cannot solve.

Here, Mamba was introduced in Gu & Dao (2023). The building block of Mamba proposed a class of selective state space models that leveraged the selection mechanism which parameterize the SSM parameters depending on the input. It additionally used a hardware-aware algorithm that computes the model recurrently with a scan not convolution. Here, Mamba overcame the issues of transformers where it showed promising results on handling data that contains long-range dependencies (LRD)s. Inspired by linear attention Katharopoulos et al. (2020), Mamba2 Dao & Gu (2024) was introduced to showcase how SSMs are now competitive and similar and transformers.

In this work, we introduce a new family of sparse SSMs based on the fundamental control theory concepts of controllability and observability developed by Kalman in 1960s. In particular, we investigate how vanilla Mamba overlooked important concepts in control theory: controllability and observability. Therefore, we propose a family of Sparse Mamba (S-Mamba) networks where a modification on the architecture of vanilla Mamba can reinforce the system to be in the controller canonical form and in the observable canonical form Bay (1999). Discussed in details in section [3.2,4], S-Mamba outperforms the original Mamba, reduces the number of parameters, and saves time in training. We start presenting our work by explaining the core structure of SSMs in Section [2]. Then, we will explain the building blocks, $(\mathbf{A}, \mathbf{B}, \mathbf{C}, \mathbf{D})$ matrices in particular, of the vanilla Mamba and our S-Mamba in Section [3]. We evaluate our work in Section [4] and present the results in Tables [1,2,3].

## 2 BACKGROUND

The purpose of this section is to dive deeper into the development and the creation of state space models (SSMs). We overview the state space equations that form the corner stone in SSMs. The train of development is then discussed to include the ideas of HiPPO matrix, LSSLs, and S4. Although the focus of the architecture and evolution on SSMs, we exclude the detailed derivations. Showcasing the parts that inspired the creation of our S-Mamba stand as the main objective.

### 2.1 STATE SPACE REPRESENTATIONS

In control theory study Hespanha (2018), researcher and scientists built their work on the fundamental state space equations. These equation can be written in the forms of Eqs. 1 and 2:

$$\dot{\mathbf{x}}(\mathbf{t}) = \mathbf{A}\mathbf{x}(\mathbf{t}) + \mathbf{B}\mathbf{u}(\mathbf{t}), \tag{1}$$

$$\mathbf{y}(\mathbf{t}) = \mathbf{C}\mathbf{x}(\mathbf{t}) + \mathbf{D}\mathbf{u}(\mathbf{t}), \tag{2}$$

- $\dot{\mathbf{x}}$: The time derivative of the state vector $\mathbf{x}$. It represents the rate of change of the state with respect to time.

- $\mathbf{x}$: The state vector, representing the internal state of the system. This vector contains all the necessary information to describe the system at a given time.

- $\mathbf{u}$: The input vector, representing external inputs or controls applied to the system.

- $\mathbf{y}$: The output vector, representing the measured or observed outputs of the system.

- $\mathbf{A}$: The state matrix, which defines how the current state $\mathbf{x}$ influences the state derivative $\dot{\mathbf{x}}$.

- $\mathbf{B}$: The input matrix, which defines how the input $\mathbf{u}$ influences the state derivative $\dot{\mathbf{x}}$.

- $\mathbf{C}$: The output matrix, which defines how the current state $\mathbf{x}$ influences the output $\mathbf{y}$.

- $\mathbf{D}$: The feed-through matrix, which defines how the input $\mathbf{u}$ directly influences the output $\mathbf{y}$.

where A is $\mathbb{R}^{n \times n}$, B is $\mathbb{R}^{n \times m}$, C is $\mathbb{R}^{p \times n}$, D is $\mathbb{R}^{p \times m}$. $n$ is the number of states, $m$ is the number of inputs, $p$ is the number of outputs.

## 2.2 HIGH-ORDER POLYNOMIAL PROJECTION OPERATOR (HiPPO)

The $HiPPO$ matrix is one the important foundations of SSMs that was proposed in Gu et al. (2020). Authors of HiPPO framework introduced a method for continuous-time memorization and can be described in Eq.(3).

$$(\text{hippo}(\mathbf{f}))(\mathbf{t}) = \text{coef}_{\mathbf{t}}(\text{proj}_{\mathbf{t}}(\mathbf{f})), \tag{3}$$

where the composition coef∘proj is called $HiPPO$. This operator is mapping a function $f : \mathbb{R}_{\geq 0} \to \mathbb{R}$ to the optimal projection coefficients $c : \mathbb{R}_{\geq 0} \to \mathbb{R}^N$.

In other words, for a continuous function $f$ at every time $t$, there is an optimal projection $g^{(t)}$ of $f$ onto the space of polynomials, with respect to a measure $\mu^{(t)}$ weighing the past. Afterwords, for an appropriately chosen basis, the corresponding coefficients $c(t) \in \mathbb{R}^N$, representing a compression of the history of $f$, satisfy linear dynamics. This continuous-time $HiPPO$ ODE can be shown in Eq.(4). The result of this will be a discretized version of the dynamics that yields an efficient closed-form recurrence for online compression of the time series $(f_k)_{k \in \mathbb{N}}$ in Eq.(5).

$$\frac{\mathbf{d}}{\mathbf{dt}}\mathbf{c}(\mathbf{t}) = \mathbf{A}(\mathbf{t})\mathbf{c}(\mathbf{t}) + \mathbf{B}(\mathbf{t})\mathbf{f}(\mathbf{t}), \tag{4}$$

$$\mathbf{c_{k+1}} = \mathbf{A_k}\mathbf{c_k} + \mathbf{B_k}\mathbf{f_k}, \tag{5}$$

for some $A(t) \in \mathbb{R}^{N \times N}$, $B(t) \in \mathbb{R}^{N \times 1}$. Where $N$ is the model size.

## 2.3 LINEAR STATE-SPACE LAYERS (LSSL)

The first attempt to build the bridge from SSMs to machine learning models was LSSLs Gu et al. (2021b), proposed by the same authors of HiPPO. Here, the linear state space layer maps the continuous in Eqs.(1), (2) to a discretized state space model $A, B, C, D$. Then, these two equations can be seen as the first view of LSSL. The discrete-time state-space model in Eqs.(6), (7) can be seen the recurrence view or the second view.

$$\mathbf{x}_t = \overline{\mathbf{A}}\mathbf{x}_{t-1} + \overline{\mathbf{B}}\mathbf{u}_t, \tag{6}$$

$$\mathbf{y}_t = \mathbf{C}\mathbf{x}_t + \mathbf{D}\mathbf{u}_t, \tag{7}$$

where the recurrent state $\mathbf{x}_{t-1} \in \mathbb{R}^{H \times N}$ carries the context of all inputs before time $t$. Then, the current state $\mathbf{x}_t$ and output $\mathbf{y}_t$ can be computed. The input $\mathbf{u} \in \mathbb{R}^{L \times H}$. $\mathbf{N}$ is the model size. $\mathbf{L}$ representing the length of a sequence where each timestep has an $\mathbf{H}$-dimensional feature vector.

The third view of LSSL is the convolution view. Then, in Eq.(8) $y$ is simply the non-circular convolution $y = K_L(\overline{A}, \overline{B}, C) * u + Du$.

$$\mathbf{y_k} = \mathbf{C}(\overline{\mathbf{A}})^k \mathbf{B} u_0 + \mathbf{C}(\overline{\mathbf{A}})^{k-1}\mathbf{B} u_1 + \cdots + \mathbf{C}\overline{\mathbf{A}}\mathbf{B} u_{k-1} + \overline{\mathbf{B}} u_k + \mathbf{D} u_k, \tag{8}$$

$$\mathbf{K_L}(\mathbf{A}, \mathbf{B}, \mathbf{C}) = \left(\mathbf{CA^i B}\right)_{\mathbf{i} \in [\mathbf{L}]} \in \mathbb{R}^{\mathbf{L}} = \left(\mathbf{CB}, \mathbf{CAB}, \ldots, \mathbf{CA^{L-1}B}\right), \tag{9}$$

where the output $y \in \mathbb{R}^{H \times L}$. $K_L$ is the Krylov function Krylov (1931).

## 2.4 STRUCTURED STATE SPACES (S4)

Creating a state model that can evolve over time to learn more information as they arrive was the main reason for creating RNNs and then LSTMs. Nevertheless, the memory remained an issue for long sequences. S4 model Gu et al. (2021a) emerged as the first SSM model built upon the concept of LSSLs. Following similar steps taken in [2.2] and [2.3], one can write the state space equations by setting the parameter $D = 0$ as it serves the purpose of a skip connection, which can be learned easily. Then, the architecture of S4 models are defined with four parameters $(\Delta, \mathbf{A}, \mathbf{B}, \mathbf{C})$.

In other words, the first step is done by taking the continuous time equations [1.2] and discritize them. Therefore, Eqs.(10) and (11) represent the recurrence view. Similarly, the convolutions view can also be rewritten as Eqs.(12) and (13).

$$\mathbf{h_t} = \overline{\mathbf{A}}\mathbf{h_{t-1}} + \overline{\mathbf{B}}\mathbf{x_t}, \tag{10}$$

$$\mathbf{y_t} = \mathbf{C}\mathbf{h_t}, \tag{11}$$

$$\overline{\mathbf{K}} = (\mathbf{C}\overline{\mathbf{B}}, \mathbf{C}\overline{\mathbf{A}}\overline{\mathbf{B}}, \ldots, \mathbf{C}\overline{\mathbf{A}}^k\overline{\mathbf{B}}, \ldots), \tag{12}$$

$$\mathbf{y} = \mathbf{x} * \overline{\mathbf{K}}, \tag{13}$$

where the transformation from parameters $(\Delta, \mathbf{A}, \mathbf{B})$ to parameters $(\overline{\mathbf{A}}, \overline{\mathbf{B}})$ is done through fixed formulas $\overline{\mathbf{A}} = f_A(\Delta, \mathbf{A})$ and $\overline{\mathbf{B}} = f_B(\Delta, \mathbf{A}, \mathbf{B})$. The pair $(f_A, f_B)$ are called discretization rule.

## 3 MAMBA

The structure of state space representations in $(\mathbf{A}, \mathbf{B}, \mathbf{C}, \mathbf{D})$ matrices has an enormous impact on the SSM's performance. Furthermore, the initialization of these matrices is as critical. Discussion in Section [2] revolved specifically around the building blocks of Mamba and around SSMs in general. Here, we present our novel method of initialization and calculation of $(\mathbf{A}, \mathbf{B}, \mathbf{C}, \mathbf{D})$ in our sparse Mamba S-Mamba. This presentation is done through showing these matrices' structure in Mamba first, then ours afterwords. From this point on, mentioning Mamba will refer to vanilla Mamba version as Mamba Gu & Dao (2023) and S-Mamba will refer to the family of sparse mamba: Controlable Mamba SC-Mamba and Observable Mamba SO-Mamba.

### 3.1 MAMBA

Building upon S4, Mamba was introduced to improve matching the modeling power of Transformers while scaling linearly in sequence length. Here, the parameter $\Delta$ in a Mamba governs how much attention is given to the current input $x_t$. It acts as a generalization of gates in Recurrent Neural Networks (RNNs). A large $\Delta$ resets the hidden state $h_t$ and focuses on the current input, while a small $\Delta$ retains the hidden state and disregards the input. This can be interpreted as a discretization of a continuous system, where a large $\Delta \to \infty$ results in the system focusing on the current input for longer, whereas a small $\Delta \to 0$ implies that the input is transient and ignored.

$$A_{nk} = \begin{cases} -\sqrt{(2n+1)(2k+1)} & \text{if } n > k, \\ -(n+1) & \text{if } n = k, \\ 0 & \text{if } n < k. \end{cases} \tag{14}$$

$$\overline{\mathbf{A}} = \exp(\Delta \mathbf{A}), \tag{15}$$

$$\overline{\mathbf{B}} = (\Delta \mathbf{A})^{-1}(\exp(\Delta \mathbf{A}) - \mathbf{I}) \cdot \Delta \mathbf{B}, \tag{16}$$

After initializing $\mathbf{A}$ based on the HiPPO matrix defined in Eq.(14), the dicretaized parameters $\mathbf{A}$ and $\mathbf{B}$ interacts with $\Delta$ through the relation of zero-order hold ($ZOH$) defined in Eqs.(15),(16) respectively. The matrices $\mathbf{B}$ and $\mathbf{C}$ in Mmaba are responsible for selectively filtering information to ensure that only relevant inputs are integrated into the state $h_t$ and subsequently into the output $y_t$. Making $\mathbf{B}$ and $\mathbf{C}$ selective allows for finer control over whether the input $x_t$ affects the state or whether the state influences the output. This selectivity enables the model to modulate its dynamics based on both the content (input) and the context (hidden states), thereby efficiently compressing a sequence model's context and discarding irrelevant information. The $\mathbf{D}$ matrix is initialized as a vector of 1's and set to be a learnable parameter as it works as a skip connection. Therefore, easy to be learned.

## 3.2 SPARSE MAMBA USING CONTROLLABLE AND OBSERVABLE FORMS

In control theory, the controllable canonical form is a specific configuration of state-space representation where the state matrix $\mathbf{A}$, the input matrix $\mathbf{B}$, and the output matrix $\mathbf{C}$ have specific structured forms. The $n \times n$ matrix $\mathbf{A}$ is arranged in such a way that it makes the system controllable Kailath (1980), and this form is particularly useful for state feedback control design discussed in section [3.2.1]. We will further implement and discuss the observable form Dullerud & Paganini (2013) in section [3.2.2].

### 3.2.1 CONTROLLABILITY

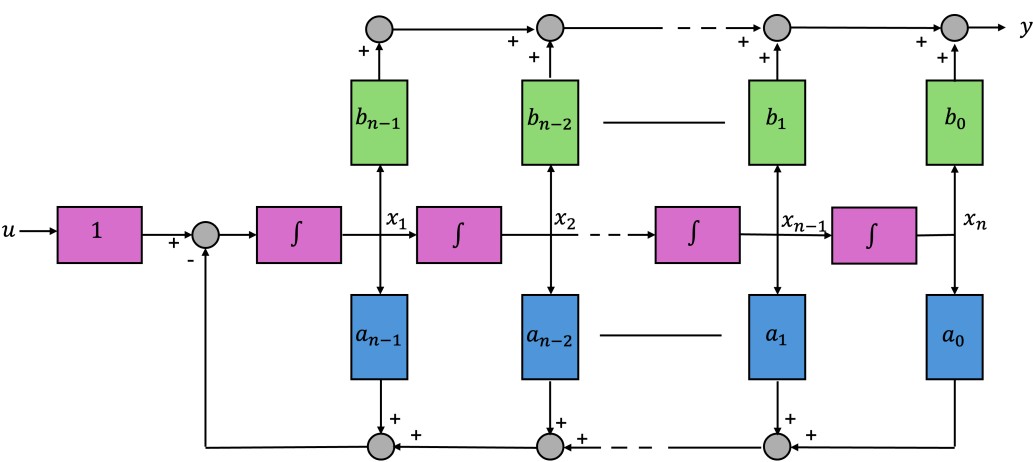

Figure 1: Block diagram analysis of controllable canonical form (CCF).

The first part of our Sparce Mamba Family is Sparce Controllable Mamba (SC-Mamba). Here, the derivation of the controllable canonical form $CCF$ is closely related to the concept of reachability. A system is said to be *reachable* if it is possible to drive the state from any initial state to any final state within a finite time interval using an appropriate control input. In other words, The system is reachable if and only if the reachability matrix $R$ has full rank Bay (1999). The CCF makes the system's controllability properties explicit. This means that it is easier to analyze and design controllers for the system because the controllability matrix is in a specific, structured form. Furthermore, since the CCF provides a clear structure, it simplifies the design of state feedback controllers. The placement of poles and zeros becomes more manageable. Here, a linear time-invariant system represented by the transfer function 17. The state matrix $\mathbf{A}$ in controllable canonical form is structured as Eq.(18).

$$H(s) = \frac{b_{n-1}s^{n-1} + b_{n-2}s^{n-2} + \cdots + b_1 s + b_0}{s^n + a_{n1}s^{n-1} + \cdots + a_1 s + a_0},\qquad(17)$$

$$\mathbf{A} = \begin{bmatrix} 0 & 1 & 0 & \cdots & 0 \\ 0 & 0 & 1 & \cdots & 0 \\ \vdots & \vdots & \vdots & \ddots & \vdots \\ 0 & 0 & 0 & \cdots & 1 \\ -a_{n-1} & -a_{n-2} & -a_{n-3} & \cdots & -a_0 \end{bmatrix}, \tag{18}$$

The input matrix $\mathbf{B}$ is a column vector, structured as:

$$\mathbf{B} = \begin{bmatrix} 0 & 0 & \cdots & 1 \end{bmatrix}^T, \tag{19}$$

The output matrix $\mathbf{C}$ in controllable canonical form can vary depending on the output structure required but is often a row vector of coefficients:

$$\mathbf{C} = \begin{bmatrix} b_{n-1} & b_{n-2} & \cdots & b_1 & b_0 \end{bmatrix}, \tag{20}$$

where $c_i$ of the transfer function. In this form, the last row of $\mathbf{A}$ contains the negatives of the transfer function coefficients $[-a_i]$ that form the characteristic polynomial of the system. We initialize $\mathbf{A}$ as a vector uniformly distributed over a given interval. Then, the vector is inserted into the controllable matrix form of $\mathbf{A}$ [18] during training. The structure of $\mathbf{B}$ [19] ensures that the input $u_t$ directly influences only the last state variable, making the system controllable from the input. The matrix $\mathbf{C}$ [20] determines how the state variables are weighted in the output $\mathbf{y_t}$, allowing selective emphasis on different state components. The $\mathbf{D}$ component in the controllable form is set to a value of $\mathbf{D} = \mathbf{0}$. However, while maintaining this setting, we set it as a learnable parameter afterward. Figure[1] is the block diagram that represents the proposed controllable structure.

Any state-space model can be converted into a controllability model by applying a similarity transformation to the state-space model that satisfies the controllable canonical form Chen (1984). Similarly, it can be converted into an observablility model, which we will describe in the next section.

### 3.2.2 OBSERVABILITY

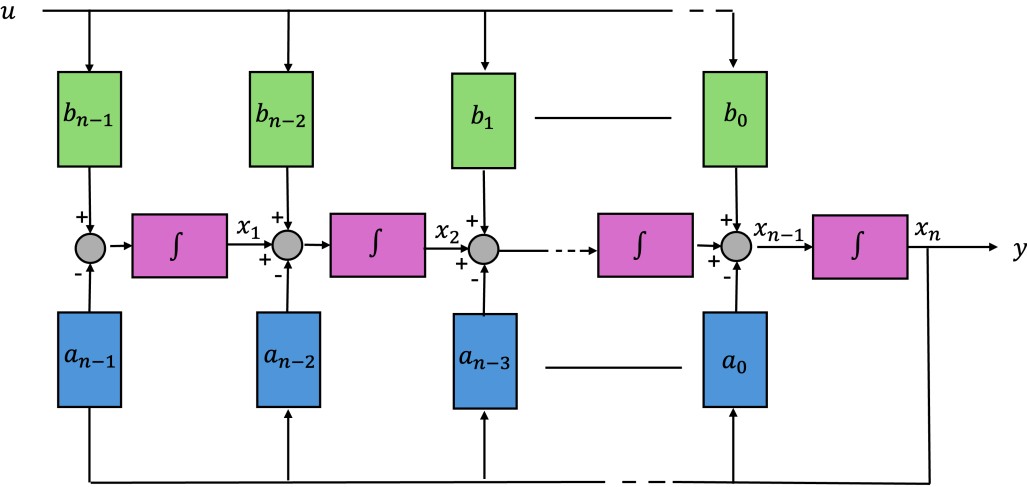

Figure 2: Block diagram analysis of observable canonical form (OCF).

The second group of Sparse Mamba's that we introduce is Sparse Observable Mamba (SO-Mamba). In this section, we will reinforce the observable canonical form OCF on the structure state space equations. Similar to CCF, the OCF makes the system's observability properties explicit. This means that it is easier to analyze and design observers for the system because the observability matrix is in a specific, structured form. Additionally, the coefficients of the characteristic polynomial of the system appear directly in the state matrix $A$. This makes it straightforward to analyze the system's dynamics and stability.

Table 1: **Perplexity Evaluation Table:** Training results comparison between vanilla Mamba, our sparse observable Mamba (SO-Mamba), and our sparse controllable Mamba ( SC-Mamba) based on perplexity matrix. Numbers in parentheses, (1M) and (100K), stand for the number of rows used in each of the datasets.

| Model | CodeParrot 1M | OpenWebText 1M | ArXiv | Cosmopidia 100K |
|-------|---------------|----------------|-------|------------------|
| Mamba | 10.4618 | 99.2525 | 70.33 | 30.5017 |
| SO-Mamba | 10.0525 | 99.3704 | 72.2705 | 30.1241 |
| **SC-Mamba** | **9.8904** | **98.5427** | **69.6179** | **30.0191** |

Table 2: **Training Time Evaluation Table:** Training results comparison between vanilla Mamba, sparse observable Mamba (SO-Mamba), and sparse controllable Mamba (SC-Mamba) based on training time. The base task in this table is the Fill-in-Middle task. Numbers in parentheses, (1M) and (100K), stand for the number of rows used in each of the datasets.

| Model | CodeParrot 1M | OpenWebText 1M | ArXiv | Cosmopidia 100K |
|-------|---------------|----------------|-------|------------------|
| Mamba | 6:27:03 | 2:27:39 | 50:32 | 36:57 |
| SO-Mamba | 6:19:08 | 2:28:26 | 51:05 | 36:43 |
| **SC-Mamba** | **6:15:39** | **2:26:11** | **50:21** | **36:32** |

The derivation of the observable canonical form is closely related to the concept of observability. A system is said to be observable if it is possible to determine the state of the system from the output measurements over a finite time interval. Here the system is observable if and only if the observability matrix $O$ has full rank Bay (1999). Therefore, one can construct the matrices in observable canonical form as:

$$\mathbf{A} = \begin{bmatrix} 0 & 0 & \cdots & 0 & -a_n \\ 1 & 0 & \cdots & 0 & -a_{n-1} \\ 0 & 1 & \cdots & 0 & -a_{n-2} \\ \vdots & \vdots & \ddots & \vdots & \vdots \\ 0 & 0 & \cdots & 1 & -a_0 \end{bmatrix}, \tag{21}$$

$$\mathbf{B} = \begin{bmatrix} b_{n-1} & b_{n-2} & \cdots & b_1 & b_0 \end{bmatrix}^T, \tag{22}$$

$$\mathbf{C} = \begin{bmatrix} 0 & 0 & 0 & \cdots & 1 \end{bmatrix}, \tag{23}$$

where the matrices in observable canonical form follow the same structures and sizes as the controllable canonical form. $\mathbf{A} \in \mathbb{R}^{n \times n}$ is an $n \times n$ matrix, the transpose of the controllable canonical form matrix. $\mathbf{B} \in \mathbb{R}^{n \times 1}$ is an $n \times 1$ column vector, the transpose of the corresponding vector in controllable canonical form. $\mathbf{C} \in \mathbb{R}^{1 \times n}$ is a $1 \times n$ row vector, the transpose of the corresponding vector in controllable canonical form. $\mathbf{D} \in \mathbb{R}$ is a scalar and will be set to be trainable. Here we can see that $\mathbf{A}$ matrix is the transpose of the controller canonical form and that $\mathbf{b}$ and $\mathbf{c}$ are the transposes of the $\mathbf{c}$ and $\mathbf{b}$ matrices, respectively, of the controller canonical form. Figure[2] is the block diagram that represents the proposed observable structure.

## 4 EXPERIMENTAL RESULTS

The first stage of our training was converting the data rows from each of the datasets and covert them into a columnar data format using LanceDB framework LanceDB (2024). We choose to prove our optimization on four popular datasets: CodeParrot Dataset [1], OpenWebText Corpus Dataset [2], On the Use of ArXiv as a Dataset [3], and Cosmopedia Dataset [4]. We indicate the count of rows used from the datasets except for $ArXiv$ dataset, where we used all available rows in the dataset.

---

[1]https://huggingface.co/codeparrot/codeparrot

[2]https://huggingface.co/datasets/Skylion007/openwebtext

[3]https://github.com/mattbierbaum/arxiv-public-datasets

[4]https://huggingface.co/datasets/HuggingFaceTB/cosmopedia

Table 3: **Number of Parameter Comparison:** The reduction of parameter analysis between Mamba, our sparse observable Mamba (SO-Mamba), and our sparse controllable Mamba (SC-Mamba) under the same settings.

| Model | Number of Parameters |
|---|---|
| Mamba | 64475648 |
| SO-Mamba | 64352904 |
| **SC-Mamba** | **64344840** |

In Table [1], we present the improvement in perplexity in our family Sparse Mamba. Here, the sparse controllable Mamba (SC-Mamba) shows an improvement of 5% in comparison to the original vanilla Mamba model. Additionally, we mention the results of enforcing the observability matrix in SO-Mamba. Table [2], shows a reduction by 3% in training time. The training for the models on each of the datasets was done for 7 epochs and the comparison was made on the last epoch.

The significant parameter reduction, presented in Table [3], demonstrates the benefits in enforcing controlability and observability in the Mamba's architecture. This reduction of 100K in parameters proves the utilization of sparsity in our family of S-Mamba. Our theoretical analysis of the architecture of Mamba2 shows that the number of parameters in our S-Mamba is also lower than the number of parameters in Mamba2.

## 5 CONCLUSION

In our paper, we introduce a family of Sparse Mamba (S-Mamba) that reinforces the controllability and observability on the original Mamba model. The controllable $n \times n$ state matrix $A$ is sparse and it has only $n$ free parameters. We showcase that our novel architecture has a better performance in terms of perplexity, less training time, and a reduction in the number of parameters in than vanilla Mamba. Our experiments prove a possibility to make any model that is based on state space representations sparse, including the diagonal structure in Mamba2, by enforcing Controllability and observability in $(\mathbf{A}, \mathbf{B}, \mathbf{C}, \mathbf{D})$ matrices. This will conclude a less complex system for language modeling using SSMs.

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
