# OpenReview forum: "Sparse Mamba: Reinforcing Controllability In Structural State Space Models"
_ICLR.cc/2025/Conference — ICLR 2025 Conference Withdrawn Submission_

### Official Review · Reviewer_ZKks · 2024-10-17

**Soundness:** 1
**Presentation:** 2
**Contribution:** 1
**Rating:** 1
**Confidence:** 5

**Summary:**

This paper proposes a study of Sparse Mamba but spends an excessive amount of space summarizing related works, such as HiPPO, LSSL, S4, and Mamba. The actual contribution of the paper seems limited to less than four pages, while the first four pages largely repeat existing material rather than introducing new insights. This raises concerns about the novelty and depth of the contribution.

Furthermore, while the paper claims to focus on sparsity, it also discusses controllability and observability, which seems tangential to the central topic. It is unusual to assess the performance of a sparse Mamba model by showing improvements in perplexity. If perplexity has indeed improved, it is unlikely to be a result of the model’s sparsity, as reducing model space typically does not enhance predictive performance. This disconnect needs clarification, as it raises questions about the actual source of the performance gains.

**Strengths:**

The paper provides a detailed introduction to HiPPO, LSSL, S4, and Mamba. However, it would be more appropriate to move this detailed background information to the appendix, as readers/reviewers are likely already familiar with these works. A single paragraph summarizing the evolution of state-space models would suffice in the main text. Additionally, these sections seem to have limited relevance to the core contribution (if any) of the paper. Shifting them to the appendix would allow for greater emphasis on the key ideas and novel aspects.

**Weaknesses:**

1. The improvement demonstrated in the paper is not significant, and it does not seem to stem from the theoretical concepts related to controllability, reachability, or observability. In fact, if the model used is Mamba, the element-wise gating mechanism should already ensure controllability. Therefore, it remains unclear what specifically accounts for the performance improvement of Sparse Mamba over the vanilla Mamba. A deeper explanation or analysis of the source of this improvement would be necessary to clarify its contribution.
2. This paper spends an excessive amount of space summarizing related works, such as HiPPO, LSSL, S4, and Mamba. The actual contribution of the paper seems limited to less than four pages, while the first four pages largely repeat existing material rather than introducing new insights. This raises concerns about the novelty and depth of the contribution. I would recommend condensing the background into a single paragraph summarizing the evolution of state-space models in the main text, while moving the detailed explanations to the appendix.

**Questions:**

1. The statement in the abstract, "However, current Mamba models lack reinforcement of controllability in state-space equations for computing the A, B, C, and D matrices at each time step, leading to increased complexity and computational costs," is not substantiated within the paper. It would be helpful if the authors provided further explanation or justification for this claim to avoid overstatement. Clarifying this point would strengthen the argument and align the abstract more closely with the content of the paper.

---

### Official Review · Reviewer_CwE7 · 2024-10-26

**Soundness:** 1
**Presentation:** 1
**Contribution:** 2
**Rating:** 3
**Confidence:** 5

**Summary:**

This paper proposes a modification to the recent state space model (SSM) Mamba. Instead of the diagonal transition matrices of previous structured SSMs such as S4 and Mamba, this paper borrows ideas from classical control theory, in particular that of controllability and observability, to propose a different class of structure transition matrices $A$: more precisely, where it has the form of a companion matrix or transposed companion matrix.

**Strengths:**

- Borrowing classical ideas from prior statistical or control work on state-based models is always a great direction to continue understanding and improving them.
- The particular form of structured matrices used and the ideas of controllability and observability make sense.

**Weaknesses:**

**Method**

The method details are very not clear. For example, one crucial detail in Mamba is that the $B$, $C$, and $A$ matrices all depend on the input. In this method, I cannot figure out if a still depends on the input (i.e. if the vector $(a_0, \dots, a_n)$ of coefficients can vary per timestep of the input sequence). Additionally, no algorithm is provided to compute the model. The exact distinctions between Mamba(-2) should be made more clear.


**Writing**
- Background reads more like a survey; there are 4 full pages of background out of an 8 page paper
- Numerous typos and formatting issues throughout
- Main figure (Figure 1) is very hard to understand and there is little description

Additionally, SpaceTime [1] should be mentioned as they were the first to use companion matrices in structured SSMs.

[1] Zhang et al. "Effectively Modeling Time Series with Simple Discrete State Spaces"

**Questions:**

The entire method description is unfortunately too vague that I cannot evaluate the technical content of the submission.

---

### Official Review · Reviewer_xdvQ · 2024-10-29

**Soundness:** 3
**Presentation:** 3
**Contribution:** 2
**Rating:** 3
**Confidence:** 3

**Summary:**

The paper aims to add controllability and observability from control theory into the Mamba SSM architecture with the hope to improve model efficiency. The model applies this to the state space matrices of the model and demonstrates these across several different corpuses.

**Strengths:**

The paper does a good job outlining the proposed changes, where they come from, and the mathematical foundations of why these changes might be considered.

**Weaknesses:**

The paper's results are extremely modest. The improvement of 3% and 5% are within the margin of error of implementations, slight training settings differences, etc... and confidence intervals are not presented, so it's not even clear this brings a benefit. The reduction in parameter count is also extremely small, 100k parameters out of 64M. It's possible the controllability and observability changes have other benefits, but there are no experiments to demonstrate this. For example, if the observability makes things more explainable, that needs to be demonstrated qualitatively and quantitatively. Additionally, the choice of datasets to evaluate are somewhat non-standard, and so its unclear why these in particular were chosen, meaning, as in perhaps they were cherry picked.

The paper needs more experimental results, more qualitative evaluation, and needs to demonstrate a stronger reason for this proposed change beyond the marginal speed improvements. As a final remark, I'm not even quite sure where the reduction in parameters comes from, as the original state matrix is sparse also only has N parameter values, even though the shape is NxN.

**Questions:**

Where does the parameter reduction come from?

---

### Official Review · Reviewer_4DXp · 2024-11-03

**Soundness:** 2
**Presentation:** 1
**Contribution:** 1
**Rating:** 3
**Confidence:** 5

**Summary:**

This paper introduces Sparse Mamba (S-Mamba), a modification of the Mamba architecture that incorporates controllability and observability structures to the pre input-varying discretization of the SSM. The authors propose two variants: Sparse Controllable Mamba (SC-Mamba) and Sparse Observable Mamba (SO-Mamba), which modify the structure of the state space matrices $A$, $B$, $C$ of the underlying continuous-time system to enforce specific canonical forms. The authors claim improvements in perplexity (5%), training time (3%), and parameter count compared to vanilla Mamba, demonstrating these results across four datasets: CodeParrot, OpenWebText, ArXiv, and Cosmopedia.

**Strengths:**

The paper makes an interesting connection between classical control theory concepts and modern state space models for machine learning, attempting to bring established theoretical frameworks to bear on neural architecture design.

* The experimental evaluation is conducted across multiple datasets of varying sizes and domains, providing some evidence for the generalizability of the approach.
* The reduction in parameter count while maintaining or improving performance is a potentially valuable contribution, as it could lead to more efficient models.

**Weaknesses:**

1. Technical Rigor and Motivation:
- The paper fails to justify why controllability and observability are desirable properties for an SSM layer in a neural network context. The authors state these properties make analysis easier, but don't explain why such analysis is necessary or beneficial for the learning task.
- The claim of "less complexity" is repeated without proper theoretical or empirical justification.
- There is no discussion of how BIBO (*bounded-input, bounded-output*) stability is maintained in the proposed architecture. This is notoriously necessary to be enforced.

2. Novelty and Literature Review:
- The paper does not acknowledge or cite previous work on using companion canonical forms in SSMs [1, 2], making it difficult to assess the novelty of the contribution.
- The motivation for making the system "sparse" is not well-explained, particularly given that the reduction in parameters (∼100K out of 64M) is relatively minor.

3. Methodology and Results:
- The paper doesn't discuss the computational implications of using non-diagonal state matrices with scan operations, which could significantly impact practical performance.
- There is no analysis of the statistical significance of the reported improvements in perplexity and training time.
- The experimental section lacks details about the inference procedure and comparison of different inference algorithms.

4. Presentation:
- The paper contains numerous issues with notation consistency (e.g., use of bold letters).
- The writing quality needs significant improvement, with unclear explanations and imprecise language throughout.
- Referenced materials are not properly used (e.g., where does the Krylov function appears in Krylov 1931?).

[1] Zhang, Michael, et al. "Effectively modeling time series with simple discrete state spaces." (2023).

[2] Parnichkun, Rom N., et al. "State-Free Inference of State-Space Models: The Transfer Function Approach." (2024).

**Questions:**

1. Could you elaborate on why controllability and observability are beneficial properties for a linear SSM layer with input-varying dynamics in a neural network? Specifically, how do could these properties improve the model's ability to learn and generalize?

2. Given that the scan operation becomes more computationally expensive with non-diagonal matrices, how does your implementation handle this trade-off? Could you provide complexity analysis comparing your approach to vanilla Mamba/Mamba2?

3. The parameter reduction achieved is relatively small compared to the total parameter count. Could you explain why this reduction is meaningful and how it affects the model's practical performance?

4. Could you provide statistical significance tests for the reported improvements in perplexity and training time? How consistent are these improvements across different random seeds and training runs?

5. How does your approach ensure stability of the learned SSM, and what impact does the canonical form constraint have on the model's expressivity?

---

### Note · Authors · 2024-11-15

I have read and agree with the venue's withdrawal policy on behalf of myself and my co-authors.